# Estimated incidence and clinical presentation of Noma in Northern Nigeria (1999–2024)

Ramat Oyebunmi Braimah[1]☯*, John Adeoye[2]☯*, Abdurrazaq Olanrewaju Taiwo[1], Seidu Bello[3], Mujtaba Bala[1], Azeez Butali[4], Bruno Oludare Ile-Ogedengbe[5], Abubakar Abdullahi Bello[6]

1 Department of Oral & Maxillofacial Surgery, Faculty of Dental Sciences, Usmanu Danfodiyo University, Sokoto, Nigeria, 2 Division of Applied Oral Sciences and Community Dental Care, Faculty of Dentistry, University of Hong Kong, Hong Kong SAR, China, 3 Cleft and Facial Deformity Foundation, Abuja, Nigeria, 4 Department of Oral Pathology, Radiology and Medicine, College of Dentistry, University of Iowa, Iowa City, Iowa, United States of America, 5 Department of Dental & Maxillofacial Surgery, Federal Medical Centre, Birnin Kebbi, Kebbi, Nigeria, 6 Noma Children's Hospital, Sokoto, Nigeria

☯ These authors contributed equally to this work.
* robdeji@yahoo.com, ramat.obraimah@udusok.edu.ng, bunmibraimah@gmail.com (ROB); jaadeoye@connect.hku.hk, jadeoye@hku.hk (JA)

## Abstract

Noma (Cancrum Oris), a recent addition to the WHO list of neglected tropical diseases, is a severe, rapidly progressing necrotizing disease of the oral cavity and facial complex with a case fatality rate of 90% if untreated. Active disease is common among children between two and six years in Sub-Saharan Africa while noma sequelae may be seen in individuals at any age. Though most cases have been reported in northern Nigeria, little research is available on the incidence of noma and its clinical presentation in this region using comprehensive data. Therefore, this study aims to estimate the incidence of noma and its clinical presentation in Northern Nigeria among different age groups. We collected retrospective data of 1,383 consecutive patients managed at Noma Children's Hospital, Sokoto, Nigeria between 1999 and 2024 for incidence estimation and description of the clinical presentation of noma. Incidence calculation was done using the WHO Oral Health Unit strategy designed with the Delphi method. Our results showed that patients were between 8 months and 80 years old with a median age (IQR) of 6 years (3–15). More patients presented with acute noma than arrested noma (67.3% vs 32.7%). The estimated incidence of noma in northern Nigeria during the study period was 87.8 cases per 100,000, with Sokoto state having the highest incidence of 691.4 cases per 100,000, while Adamawa state had the lowest incidence of 11.2 cases per 100,000. The annual average and median incidence of noma across all years was 3.4 and 1.6 cases per 100,000 (range: 0.2-16.6 cases per 100,000), although between 2020 and 2024, the annual average and median incidence estimates were 12.0 and 12.6 cases per 100,000. Also, this study found the incidence of noma cases with gangrene to be higher than cases with oedema or acute necrotizing ulcerative gingivitis. These findings confirm the high incidence and impact

**Data availability statement:** Data used for this study is has been deposited in a public repository available at: https://doi.org/10.5281/zenodo.14554768

**Funding:** The author(s) received no specific funding for this work.

**Competing interests:** The authors have declared that no competing interests exist.

of noma in northern Nigeria in the last two and half decades and highlight the need to intensify awareness of risk factors and early signs of noma within communities in the region and to conduct community-based screening to promote the identification and cost-effective treatment of reversible early noma disease.

## Author summary

Noma is a severe and deadly gangrenous disease that affects the mouth and other tissues of the face. It is often associated with poverty and is currently prevalent in Sub-Saharan Africa. More cases have been described in northern Nigeria than in any African region. However, the incidence of noma and its clinical presentation in this region is unknown from data available from different states. In this study, we retrospectively collected data from the only dedicated specialist hospital for noma intervention in Nigeria between 1999 and 2024 to calculate noma incidence within this period. Then the incidence of noma was calculated using a method proposed by the World Health Organization (WHO). This study estimates an incidence of 87.8 cases per 100,000 within the study period (26 years) and an annual average and median incidence of 3.4 and 1.6 cases per 100,000. Of note, the incidence of noma with gangrene (stage 3) was higher than the incidence of oedema (stage 2) or acute necrotizing ulcerative gingivitis (stage 1). These results highlight the importance and impact of noma in northern Nigeria and underscore the need for noma awareness and screening programs for population education and detection of early noma disease which are reversible with cost-effective interventions.

## Introduction

Noma is a devastating and disfiguring necrotizing disease that primarily affects malnourished children in low-income countries or areas with extreme poverty. With a reported mortality rate exceeding 90% in untreated cases, noma has been recognized as a public health challenge for many countries in sub-Saharan Africa [1]. Acute noma predominantly affects children between the ages of two and six years old in poorly developed countries where adequate nutrition, sanitation, and cleanliness are lacking [1]. On the contrary, noma sequelae which include scars and facial defects may be seen at any age. Nonetheless, a few acute noma cases have been reported in northern Nigeria among adult patients [2]. Despite its severe impact, the aetiology of noma remains poorly understood, which limits effective preventive and therapeutic strategies [3]. Historically, noma was associated with a combination of malnutrition, poor oral hygiene, measles, and bacterial infections [3,4]. However, recent evidence suggests a multifactorial origin involving complex interactions possibly between genetic predisposition, host immune response, environmental factors which may compromise patient's immune system resulting in microbial dysbiosis and noma susceptibility [5–7].

The geographical distribution and prevalence of noma continue to generate major questions globally because very little data exists about its incidence, mortality rate, and other epidemiological factors. Similarly, reasons for noticeable dissimilarities in the incidence and prevalence among comparable population groups of different but equally impoverished and underdeveloped countries remain vague [8,9]. To emphasize research that measure the efficacy of any preventive intervention by policymakers, it is important to record and properly understand the epidemiology of the disease condition [10]. Therefore, this present study aimed to estimate the incidence and determine the clinical presentation of noma in northern Nigeria.

## Methods

### Ethics

Approval to conduct this study was granted by the Sokoto State Ministry of Health Ethics Research Committee (Registration no: SKHREC/037/2024). Informed consent was waived due to the retrospective nature of this study. All potential patient identifiers were removed from the data collected, and anonymized data was used for analyses.

This retrospective study involved the health records of consecutive patients with noma managed at the Noma Children's Hospital (NCH), Sokoto Nigeria, from September 1999 to October 2024. Founded in 1999, NCH is the only specialized health facility in Nigeria, and one of few worldwide, dedicated to managing acute noma and noma survivors. The model of care at the hospital involved the provision of intensive care during active disease, management of noma sequelae, multidisciplinary care for patients with noma, and community-based services [11].

### Data collection

Eligible patients for data extraction from medical records were those diagnosed with noma (active disease and post-disease defects) at NCH within the study period who resided in any of the nineteen northern Nigerian states (Adamawa, Bauchi, Benue, Borno, Gombe, Jigawa, Kaduna, Kano, Katsina, Kebbi, Kogi, Kwara, Nasarawa, Niger, Plateau, Sokoto, Taraba, Yobe, Zamfara) and Federal Capital Territory at the time of diagnosis. The rationale for limiting data collection to Northern Nigeria was due to the predominant number of noma cases from this region [12–14] and NCH's location in the country. Individuals across all age groups were included. However, patients with necrosis or gangrene affecting anatomic sites outside the orofacial region were not considered. Also, we excluded records of patients who resided outside Nigeria at the time of diagnosis, and patients with missing data on their state of residence at diagnosis were excluded.

Patient information was collected using an electronic spreadsheet, and variables collected included demographic data (i.e., age and sex), year of diagnosis, state of origin, and state of residence at diagnosis. Also, the orofacial sites affected by noma, body weight at diagnosis, and hemoglobin concentration were collected. Patients were also categorized according to the five stages of disease progression recommended by the World Health Organization (WHO) i.e., acute necrotizing ulcerative gingivitis (ANUG; 1), oedema (2), gangrene (3), scarring (4), and sequelae (5). Stages 1 and 2 are reversible while stages 3–5 are irreversible stages. However, for the purpose of this study, persons with "acute noma" referred to those that had noma stages 1–3 at presentation while "arrested noma" was used as a collective term for patients with noma stages 4 and 5 at presentation.

### Statistical analysis

Data was analyzed using Statistical Package for the Social Sciences (SPSS) v 29 (IBM Corp, Armonk, NY, USA). Descriptive analysis was performed for categorical and continuous variables and presented as texts, tables, and figures. The normal distribution of continuous variables was also determined using Kolmogorov-Smirnov's test before significant differences (or otherwise) by categories were determined using Mann-Whitney U test (for two categories) or Kruskal-Wallis test (for three or more categories). Pearson's Chi-square test was conducted to determine significant differences in

the proportion of categorical variables if relevant statistical assumptions were met. Else, Fisher's exact test was used. For all tests, probability values below 5% were considered statistically significant.

This study estimated the incidence of noma in the study period considering only patients with acute disease. Incidence analysis was done following the WHO Oral Health Unit's 1994 expert consultation report conducted using the Delphi method as a two-step process [12,15,16]. Also, only data from patients with acute noma (WHO stages 1–3) were used for incidence calculation. Though two states (Nasarawa and Plateau) had no cases with acute noma in the dataset, their population was included for incidence estimation of noma in Northern Nigeria. First, the total number of surviving cases (S) was determined as a ratio of the number of cases referred to NCH (provided by our data, R) and the estimated percentage of surviving cases that were referred (x). This is given by the equation:

$$S = \frac{R * 100}{x}$$

where x was determined adaptively based on the distance of each state of residence at diagnosis to NCH, which may affect patients' frequency of presentation to the treatment center. Based on previous reports [12,15] and expert opinions, we first approximated that the proportion of surviving cases referred to NCH within Sokoto state was 20%. Then, for every 100 km distance of the states from Sokoto, we estimated that the proportion of surviving cases referred decreased by 1% based on experts' opinion. As such, different values of x were used for incidence calculation as shown in Table 1.

The incidence of noma (I) was then calculated for each state as the ratio of the number of surviving cases (S) and the case survival rate of noma (y) which was 10% according to previous studies [15,17,18].

$$I = \frac{S * 100}{y}$$

The number of new cases calculated at the state level were totaled before incidence estimation at the regional level (i.e., overall incidence). The incidence of noma was also determined for each age category and sex. The 2006 Nigerian Population Census results is the latest population estimate available for our population and served as the reference population used for reporting incidence values in this study [19]. For estimated incidence calculation per age group or sex, we

**Table 1. Proportion of surviving cases referred to NCH used to calculate number of surviving cases.**

| States | Average Distance to Sokoto (km) | Geopolitical zone | Distance rank | Proportion of surviving cases, x (%) |
|---|---|---|---|---|
| Adamawa | 1156.2 | Northeast | 13 | 8.44 |
| Bauchi | 823 | Northeast | 10 | 11.77 |
| Borno | 1080.5 | Northeast | 12 | 9.2 |
| Jigawa | 650.5 | Northwest | 8 | 13.49 |
| Kaduna | 542 | Northwest | 6 | 14.58 |
| Kano | 535 | Northwest | 5 | 14.65 |
| Katsina | 362 | Northwest | 3 | 16.38 |
| Kebbi | 150.9 | Northwest | 1 | 18.49 |
| Nasarawa | 837 | Northcentral | 11 | 11.63 |
| Niger | 497.6 | Northcentral | 4 | 15.02 |
| Plateau | 628 | Northcentral | 7 | 13.72 |
| Sokoto | 0 | Northwest | 0 | 20 |
| Yobe | 676 | Northeast | 9 | 13.24 |
| Zamfara | 232.5 | Northwest | 2 | 17.68 |

also used the specific reference population for that age group and sex (i.e., population at risk). Due to the hospital-based dataset employed, this study did not perform prevalence analysis since NCH data only accounts for a fraction of the expected total cases (especially for patients with post-noma defects) that will not be considered if prevalence calculation was performed. Incidence mapping was done using GeoDa v 1.20 [20]. Map shapefiles and boundaries were obtained from the Humanitarian Data Exchange platform of the UN Office for the Coordination of Humanitarian Affairs (OCHA), Nigeria [21].

## Results

### Patient demographics

One thousand three hundred and eighty-three patients with noma managed at NCH between 1999 and 2024 were included for analysis in this study. Detailed description of patients and their demographic characteristics are presented in Table 2. Patients were between 8 months and 80 years old with a median age (IQR) of 6 years (3–15). Overall, more patients were below 5 years old (n = 569, 44.1%), while the proportions of children between five and nine years old, adolescents (10–17 years), young adults (18–39 years), middle-aged adults (40–64 years), and elderly patients (≥65 years) were 24.5% (n = 339), 11.9% (n = 165), 16.3% (n = 225), 5.2% (n = 72), and 0.9% (n = 13) respectively. Also, most

**Table 2.** Demographic distribution of 1383 noma cases managed at NCH, Sokoto between 1999 to 2024 at presentation.

| Variables | | Active disease (n = 931) | Arrested noma (n = 452) | All cases (%) | p-value |
|---|---|---|---|---|---|
| Age | Median (IQR) | 4 (3 − 7) | 20 (9 − 31.8) | 6 (3 − 15) | <0.001[a] |
| | Median (95% CI) | 4 (4 − 4) | 20 (18 − 23) | 6 (5 − 6) | |
| Age (category) | < 5 years | 523 (56.2) | 46 (10.2) | 569 (41.1) | <0.001[b] |
| | 5 − 9 years | 262 (28.1) | 77 (17.0) | 339 (24.5) | |
| | 10 − 17 years | 89 (9.6) | 76 (16.8) | 165 (11.9) | |
| | 18 − 39 years | 52 (5.6) | 173 (38.3) | 225 (16.3) | |
| | 40 − 64 years | 5 (0.5) | 67 (14.8) | 72 (5.2) | |
| | > 65 years | 0 | 13 (2.9) | 13 (0.9) | |
| Sex | Female | 430 (46.2) | 203 (44.9) | 633 (45.8) | 0.655[b] |
| | Male | 501 (53.8) | 249 (55.1) | 750 (54.2) | |
| State of residence at diagnosis | Adamawa | 3 (0.3) | 7 (1.5) | 10 (0.7) | <0.001[b] |
| | Bauchi | 18 (1.9) | 9 (2.0) | 27 (2.0) | |
| | Borno | 7 (0.8) | 3 (0.7) | 10 (0.7) | |
| | Jigawa | 14 (1.5) | 7 (1.5) | 21 (1.5) | |
| | Kaduna | 21 (2.3) | 8 (1.8) | 29 (2.1) | |
| | Kano | 59 (6.3) | 78 (17.3) | 137 (9.9) | |
| | Katsina | 28 (3.0) | 6 (1.3) | 34(2.5) | |
| | Kebbi | 111 (11.9) | 59 (13.1) | 170 (12.3) | |
| | Nasarawa | 0 | 1 (0.2) | 1 (0.1) | |
| | Niger | 9 (1.0) | 10 (2.2) | 19 (1.4) | |
| | Plateau | 0 | 5 (1.1) | 5 (0.4) | |
| | Sokoto | 512 (55.0) | 197 (43.6) | 709 (51.3) | |
| | Yobe | 29 (3.1) | 9 (2.0) | 38 (2.7) | |
| | Zamfara | 120 (12.9) | 53 (11.7) | 173 (12.5) | |

[a]Mann-Whitney U test;

[b]Pearson Chi-square test/Fisher's exact test.

*Comparisons was between patients with active disease and arrested noma at presentation based on the demographic variables.

patients with noma presenting to NCH were males (n = 750, 54.2%) than females (n = 633, 45.8%). According to the state of residence, most patients were from Sokoto state (n = 709, 51.3%), followed by Zamfara (n = 173, 12.5%), Kebbi (n = 170, 12.3%), and Kano state (n = 137, 9.9%). More noma cases presented to NCH in 2021 (n = 220, 15.9%) than in other years, with patients between 2021 and 2024 accounting for 51.7% of all cases (n = 714) (Fig 1). When the year of diagnosis was stratified by the state of residence, the analysis showed that Sokoto state had the highest number of cases in 23 of 26 years considered in this study (range: 3 cases in 2009–116 cases in 2021; p < 0.001) (Fig 2). Kano state had the highest number of cases in 2000, while Kebbi state had the highest number in 2019.

### WHO staging and clinical presentation

More individuals presented with active noma disease (i.e., stages 1–3) than arrested noma (i.e., stages 4 and 5) (n = 931, 67.3% vs n = 452, 32.7%) (Table 2). Upon stratifying the active noma disease by the three pertinent WHO stages (i.e., stages 1–3), the analysis showed that most patients presented to NCH with gangrene (n = 825, 59.7%) than oedema (n = 101, 7.3%), or ANUG (n = 5, 0.4%). Likewise, for arrested noma (i.e., stages 4 and 5), more patients have scarring (n = 346, 25.0) than sequelae of acute noma (n = 106, 7.7%) (Table 3). Cases with arrested noma had a significantly higher median age than cases with active disease (20 years vs 4 years old, p < 0.001). Following age stratification, this study also found that a significantly higher proportion of patients with active disease were below five years old (n = 523, 56.2%) while more patients with arrested noma were between 18 and 39 years old (p < 0.001). All patients that were 65 years and above had post-active disease defects (n = 13, 2.9%) (p < 0.001). A detailed analysis of the patient's age and the five noma stages is also shown in Table 3. However, a similar proportion of males and females had acute and arrested noma (p = 0.655, Table 2). This study also found that a significantly higher proportion of cases presenting to the hospital from Sokoto state were acute noma (55% vs 43.6%), while more cases with arrested noma were from Kano (17.3% vs 6.3%) and Kebbi states (13.1% vs 11.9%) (p < 0.001, Table 2).

The median body weight (IQR) of all noma cases at diagnosis was 15 kg (9.3-41). Based on the disease category, patients with acute noma had a median weight of 11 kg while patients with arrested noma had a median weight of 45.5 kg (p < 0.001; Table 4). Detailed stratification of the median body weights by different age groups is shown in Table 3. This ranged from 8.8 kg (7.1-10.6) among children below five years to 55.5 kg (48-62.3) among cases who were between 40 and 64 years old (i.e., middle age group). The median hemoglobin concentration (IQR) at diagnosis for all cases was 11.1 g/dL (9.1-12.6), ranging from 7.8 g/dL (7.8 – 9.5) among elderly cases (above 65 years old) to 13.1 g/dL (11.4-14.8) among young adults (18–39 years old) (Table 3).

Analysis based on the sites affected by noma showed that the right cheek was affected in 55.5% (n = 768) of patients, while the left cheek was affected in 49.4% of patients (n = 683). The lower lip was involved in 1.8% of patients (n = 25) while 1.3% (n = 18) of patients had noma disease that affected the nose and eyes. Also, upper lip involvement was observed among 1.2% of patients (n = 17). Of note, significantly more patients below five years old had diseases that affected the nose and both lips while more patients who had noma that affected the right cheek were above ten years old (p < 0.001-0.012). This was also in line with noma disease category as significantly more patients with noma affecting the nose and lip had active disease on presentation (p < 0.001 – 0.004, Table 4).

### Incidence of Noma

The estimated total number of new cases of noma in Northern Nigeria between 1999 and 2024 based on our NCH data was 52,041, with an overall incidence of 87.8 cases per 100,000 population. The estimated incidence of noma among males within 26 years in Northern Nigeria was 91 cases per 100,000 population (of males), while 84.4 cases per 100,000 population (of females) was calculated for females. Fig 3 showed the 26-year estimated incidence of noma by the different states in Northern Nigeria, with Sokoto state having the highest incidence of 691.4 cases per 100,000 population and Adamawa state having the lowest incidence of 11.2 cases per 100,000 population. Within the study period, the estimated

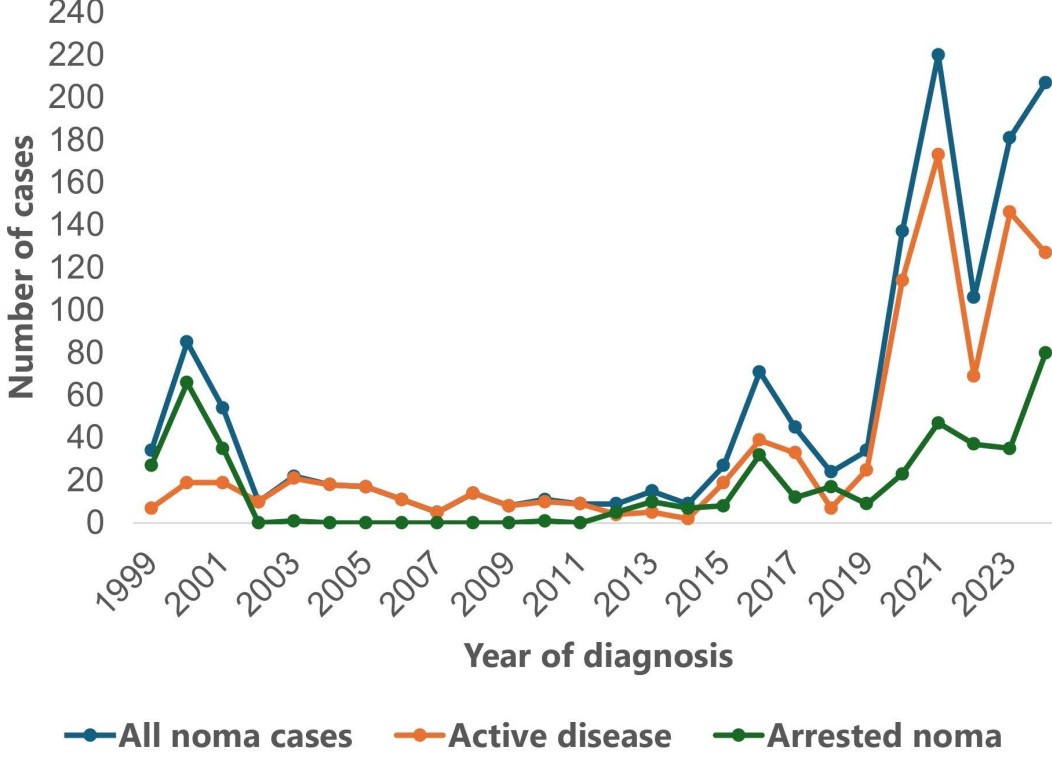

**Fig 1. Line plot showing the annual presentation of noma cases to NCH, Sokoto between 1999 and 2004.**

annual incidence of noma ranged from 0.2 cases per 100,000 population in 2014 to 16.6 cases per 100,000 in 2021 (Fig 4). Notably, this study observed that the estimated annual incidence of noma increased significantly between 2020 and 2024 (6.6 cases per 100,000 population to 16.6 cases per 100,000 population) compared to other periods (Fig 4). The average and median estimated annual incidence values across all years was 3.4 and 1.6 cases per 100,000, while the average and median estimated annual incidence between 2020 and 2024 was 12.0 and 12.6 cases per 100,000 respectively.

When stratified by the WHO noma stages 1–3, this study found the estimated 26-year incidence of ANUG to be 0.4 cases per 100,000, 9.1 cases per 100,000 for patients with oedema, and 78.3 cases per 100,000 population for patients with gangrene. Compared to other age groups, the estimated 26-year noma incidence was higher among patients below ten years old, i.e., 297.9 cases per 100,000 population at risk among children below five years and 181.7 cases per 100,000 population at risk among children between five and nine years old (Fig 5). Among individuals between 0 and 17 years, the estimated 26-year incidence of noma was higher in Sokoto state than in other states within Northern Nigeria (287.2 to 2870.3 cases per 100,000 population at risk, Fig 6). However, the estimated 26-year noma incidence among young adults was higher in Kebbi state (104.4 cases per 100,000 population at risk) compared to other states studied. Also, new noma cases among middle-aged adults were only recorded in Kebbi state within the study period (26-year incidence: 58.9 cases per 100,000 population at risk, Fig 6).

## Discussion

Noma, the most recent addition to the WHO's Neglected Tropical Diseases list, is a severe necrotizing disease of the orofacial complex with a case fatality rate of 90%. It often affects children between two and six years within Sub-Saharan

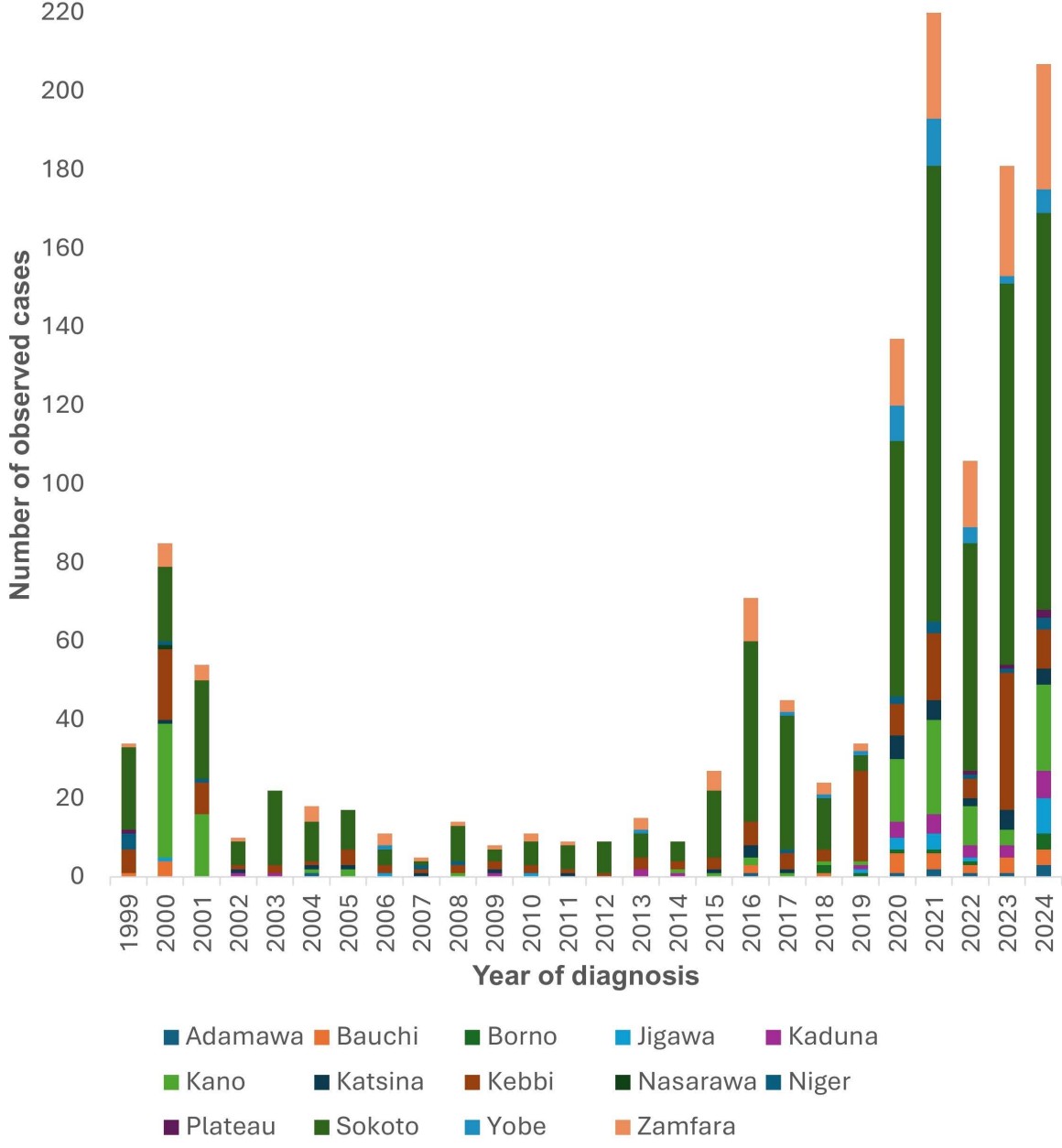

**Fig 2. Stacked bar plot showing the number of noma cases presenting to NCH, Sokoto between 1999 and 2024 by their state of residence at diagnosis.**

Africa, and most cases from this region in the last three decades have been reported in northern Nigeria [14,22]. Given the scarcity of population data on noma incidence due to its association with low-income populations, this study employed data from a large specialist hospital dedicated to noma prevention and management to estimate the incidence of noma (from 1999 to 2024) in Northern Nigeria.

This is the first study to estimate the incidence of noma and describe its clinical presentation across different states within Northern Nigeria. Our analysis estimated the incidence of noma in northern Nigeria, from 1999 to 2024, is 87.8 cases per 100,000, with a slightly higher incidence of 91 cases per 100,000 among males and 84.4 cases per 100,000

**Table 3. Clinical stage and presentation of 1383 noma cases managed at NCH between 1999 to 2024.**

| Variables | | Age of patients | | | | | | All cases | p-value |
|---|---|---|---|---|---|---|---|---|---|
| | | <5 years | 5–9 years | 10–17 years | 18–39 years | 40–64 years | >65 years | | |
| WHO noma stage (n = 1383) | 1 | 3 (0.5) | 2 (0.6) | 0 | 0 | 0 | 0 | 5 (0.4) | <0.001[a] |
| | 2 | 81 (14.2) | 11 (3.2) | 6 (3.6) | 3 (1.3) | 0 | 0 | 101 (7.3) | |
| | 3 | 439 (77.2) | 249 (73.5) | 83 (50.3) | 49 (21.8) | 5 (6.9) | 0 | 825 (59.7) | |
| | 4 | 42 (7.4) | 72 (21.2) | 73 (44.2) | 124 (55.1) | 28 (38.9) | 7 (53.8) | 346 (25.0) | |
| | 5 | 4 (0.7) | 5 (1.5) | 3 (1.8) | 49 (21.8) | 39 (54.2) | 6 (46.2) | 106 (7.7) | |
| Body weight (kg) (n = 858) | Median (IQR) | 8.8 (7.1-10.6) | 15 (11.1-19) | 31 (20-38.3) | 52 (45-60) | 55.5 (48-62.3) | 42.5 (36-67) | 15 (9.3 – 41) | <0.001[b] |
| Hemoglobin concentration (g/dL) (n = 921) | Median (IQR) | 9.3 (7.4-11.2) | 11 (9.6-12.1) | 12.1 (10 – 13.1) | 13.1 (11.4-14.8) | 12.2 (11.5 – 14) | 7.8 (7.8-9.5) | 11.1 (9.1 – 12.6) | <0.001[b] |
| Anatomic sites affected (n = 1383) | Nose | 15 (2.6) | 3 (0.9) | 0 | 0 | 0 | 0 | 18 (1.3) | 0.012[a] |
| | Right cheek | 272 (47.8) | 187 (55.2) | 100 (60.6) | 154 (68.4) | 46 (63.9) | 9 (69.2) | 768 (55.5) | 0.012[a] |
| | Left cheek | 283 (49.7) | 167 (49.3) | 89 (53.9) | 106 (47.1) | 34 (47.2) | 4 (30.8) | 683 (49.4) | 0.578[a] |
| | Upper lip | 17 (3.0) | 0 | 0 | 0 | 0 | 0 | 17 (1.2) | <0.001[a] |
| | Lower lip | 21 (3.7) | 4 (1.2) | 0 | 0 | 0 | 0 | 25 (1.8) | <0.001[a] |
| | Eye | 7 (1.2) | 3 (0.9) | 2 (1.2) | 5 (2.2) | 1 (1.4) | 0 | 18 (1.3) | 0.828[a] |

[a]Pearson's Chi-square test/Fisher's exact test;

[b]Kruskal-Wallis test.

among females living in the region. Moreover, the estimated incidence of noma was found to vary between 0.2-16.6 cases per 100,000 annually, with Sokoto and Adamawa states having the most and least incidence respectively. This study also found that the overall incidence of noma was generally higher among patients below 10 years old. However, 26-year incidence estimates among persons between 18 and 64 years old were higher in Kebbi state compared to other areas within northern Nigeria.

The incidence of noma in the northern Nigeria subregions has been investigated in different studies [12,22,23]. Recently, using twelve acute noma cases encountered between 2010 and 2018, Bello et al estimated the incidence of noma within north-central Nigeria to be 8.3 cases per 100,000 population, ranging from 4.1 to 17.9 cases per 100,000 in the different states that make up the subregion [12]. In this study, we estimated that the incidence of noma in northern

**Table 4. Clinical presentation of 1383 noma cases managed at NCH between 1999 to 2024 by disease stage category.**

| Variables | | Active disease (n = 931) | Arrested noma (n = 452) | p-value |
|---|---|---|---|---|
| Body weight (kg) (n = 858) | | 11 (8.45-18) | 45.5 (20.43-55.5) | <0.001[a] |
| Hemoglobin concentration (g/dL) (n = 921) | | 10.5 (8.7-12.1) | 11.8 (10.3-13.3) | <0.001[a] |
| Anatomic sites affected (n = 1383) | Nose | 18 (1.9) | 0 | 0.003[b] |
| | Right cheek | 496 (53.3) | 272 (60.2) | 0.015[b] |
| | Left cheek | 419 (45.0) | 264 (58.4) | <0.001[b] |
| | Upper lip | 17 (1.8) | 0 | 0.004[b] |
| | Lower lip | 25 (2.7) | 0 | <0.001[b] |
| | Eye | 3 (0.3) | 15 (3.3) | <0.001[b] |

[a]Mann-Whitney U test;

[b]Pearson Chi-square test.

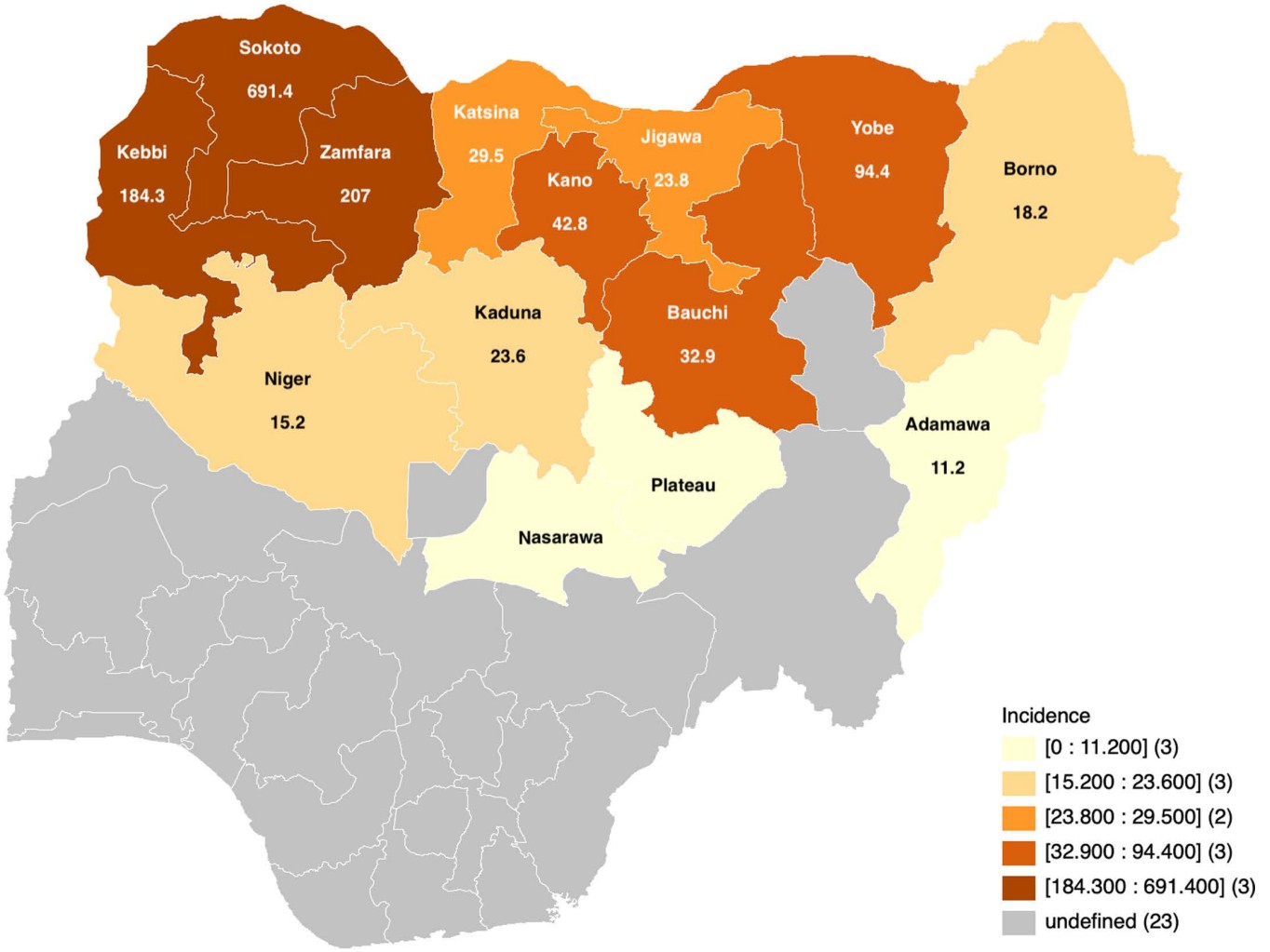

**Fig 3. Map of Nigeria showing the estimated incidence of noma (per 100,000 population) in different states in Northern Nigeria from 1999 to 2024 (Shapefile source:** https://data.humdata.org/dataset/cod-ab-nga**).**

Nigeria is 87.8 cases per 100,000 population, which comparatively, is about tenfold higher than the incidence estimated for north-central Nigeria alone. Though similar methods were used to estimate the incidence of noma in Bello et al's study and ours, this finding may be attributed to the wider regional scope and study period considered in this study [12]. In Sokoto state (northwest Nigeria), Fieger et al also analyzed 378 noma cases and estimated an average incidence of 640 cases per 100,000 among persons aged 10–30 between 1996 and 2001 (range: 440–850 cases per 100,000 population) [23]. In this study, the incidence of noma among persons between 10 and 40 years in Sokoto ranged between 80.4 and 287.2 cases per 100,000, which is lower than the estimates reported by Fieger et al. This finding may be related to the different approaches for incidence estimation in both studies, with Fieger et al calculating noma incidence using the incidence of cleft lip, age, location of the patients relative to the treatment center, and noma mortality. Further comparing our calculated incidence with those reported by Lafferty between 2002 and 2012 [24], this study finds that the incidence of noma in northern Nigeria is about fifteen folds higher than reported incidences of noma in Ethiopia among children between 0 and 9 years old. Regarding the annual estimated incidence of noma, our study showed a significant increase in

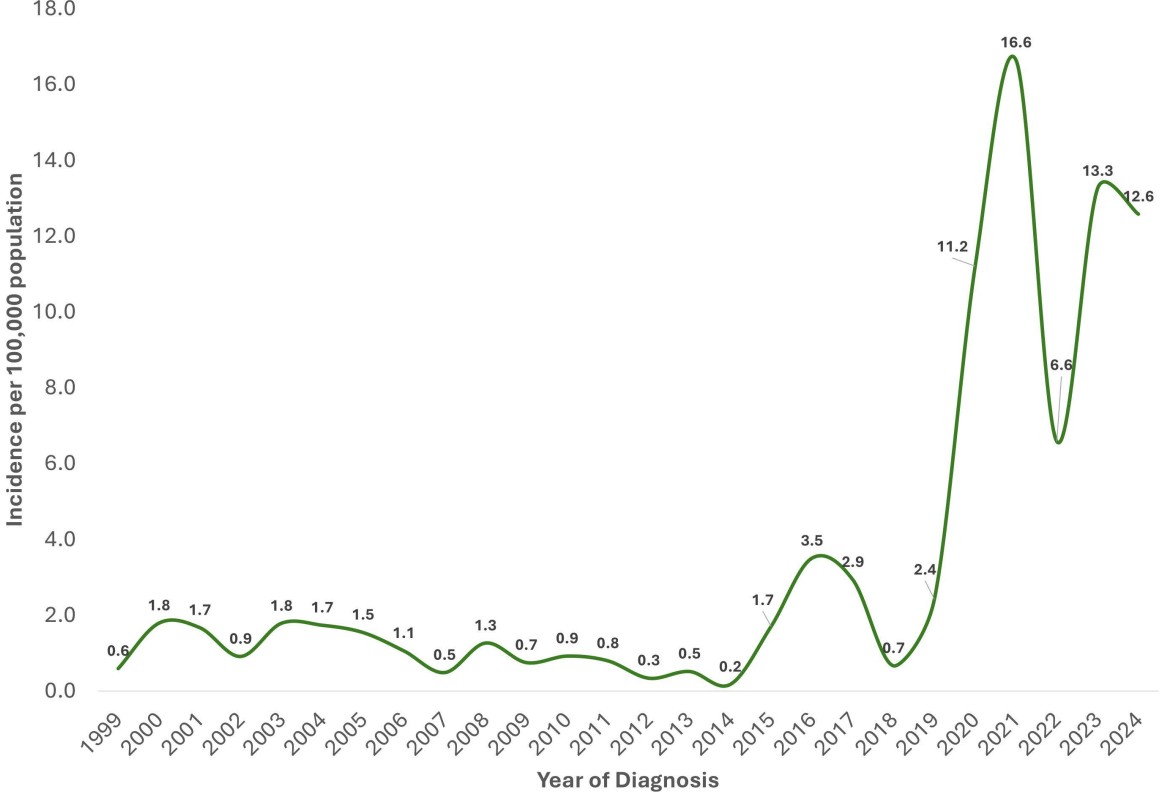

**Fig 4. Estimated noma incidence in Northern Nigeria by year of diagnosis from 1999 to 2024.**

number of new cases between 2020 and 2024 compared to previous years. This may be as a result of increased awareness and sensitization on noma in recent years among rural communities by governmental and non-governmental stakeholders like Médecins Sans Frontières resulting in increased case presentation [11].

This study found that the incidence of noma was higher in Sokoto than in other states within Northern Nigeria. This finding supports the systematic review by Galli and colleagues [22], highlighting that Sokoto was the subnational region with the highest number of new noma cases in Africa. This finding is expected given that Sokoto state has the highest poverty rate in Nigeria [25]. Nonetheless, this finding may also be due to the location of NCH and higher case presentation from Sokoto residents than other parts in Northern Nigeria which may also explain the clustering of high noma incidence in Sokoto, Kebbi, and Zamfara states due to proximity. However, our analysis showed that the high case incidence in Sokoto notion holds only for patients below 18 years, but not young and middle-aged adults with acute noma, where the highest incidence during the study period was observed in Kebbi state. Noma in adults is usually associated with immunocompromising conditions like HIV infection/AIDS and leukemia, and this study's finding may be related to the higher HIV prevalence in Kebbi than in Sokoto state [26].

Our findings on the pattern of acute noma presentation and staging of noma in northern Nigeria showed that there were significantly more patients with gangrene (irreversible stage) than oedema or ANUG (reversible stages) during the study period. This further supports that noma presentation in Northern Nigeria is frequently in the later stages of the disease which may be due to lack of awareness of early disease signs or poor access to health centers for early treatment. As such, there is a need to intensify efforts aimed at creating awareness of the early signs of noma in the region, providing nutritional support and oral health sensitization for at-risk communities, and screening children below ten years to identify

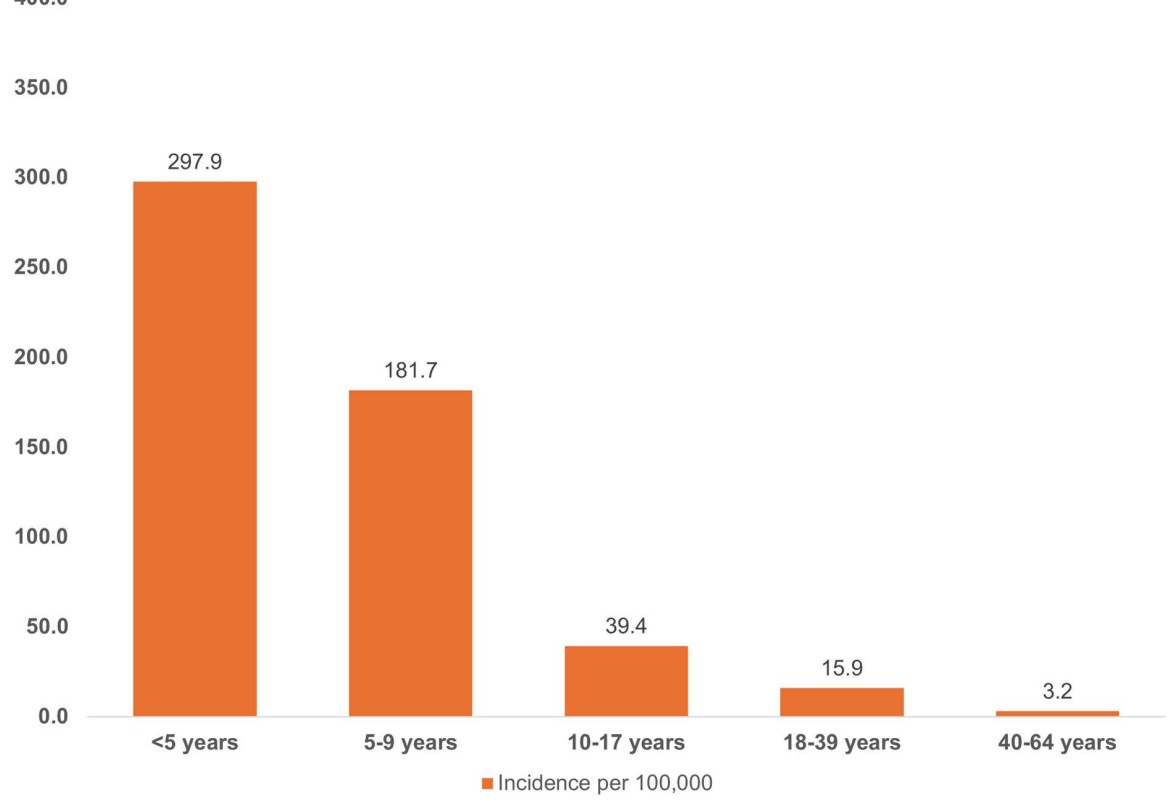

**Fig 5. Bar plot showing estimated noma incidence (26 years) among different age groups in Northern Nigeria.**

those with simple gingivitis, ANUG, and oedema for referral and management. In this study, noma mostly involved the cheeks, followed by the lower lip, nose, and eyes. Moreover, patients with acute disease exclusively had lesions involving the nose and/or lips. This finding is similar to those of Adeniyi and Awosan [27], who used a subgroup of the data employed in this study to describe the pattern of noma in northwest Nigeria. However, Bello et al [12] found that noma in northcentral Nigeria often affected the nose and upper lip than other sites, which is in contrast to our observation on the affected sites.

Though our study uniquely estimated noma incidence in Northern Nigeria from 1999 to 2024, with the largest cohort of noma cases to date, it is not without limitations. The main limitation is in the use of hospital-based data from a single center for regional noma incidence estimation. Though our approach tried to adjust for proportion of patients presenting to the hospital, distance from hospital, and surviving case proportions, the demographic profile and clinical presentation of patients that suffered mortality or did not present to NCH may still be significantly different from the cohort used for analysis in this study. As such, the incidence estimates and case demographics in this study should be interpreted given this limitation. Another limitation is the lack of data from six out of 20 eligible northern Nigeria states/territories, i.e., Federal Capital Territory, Gombe, Taraba, Benue, Kwara, and Kogi. Nonetheless, the latter three states have the lowest poverty rates in northern Nigeria, which may indicate a low incidence/prevalence of noma in these areas. Third, little is known about the number of noma cases from Gombe or Taraba to date, which remains to be explored in future studies. Last, given that a retrospective cohort was used for analysis, this data used in this study did not allow for detailed analysis of noma risk factors or rationale for case presentation or demographics in different areas.

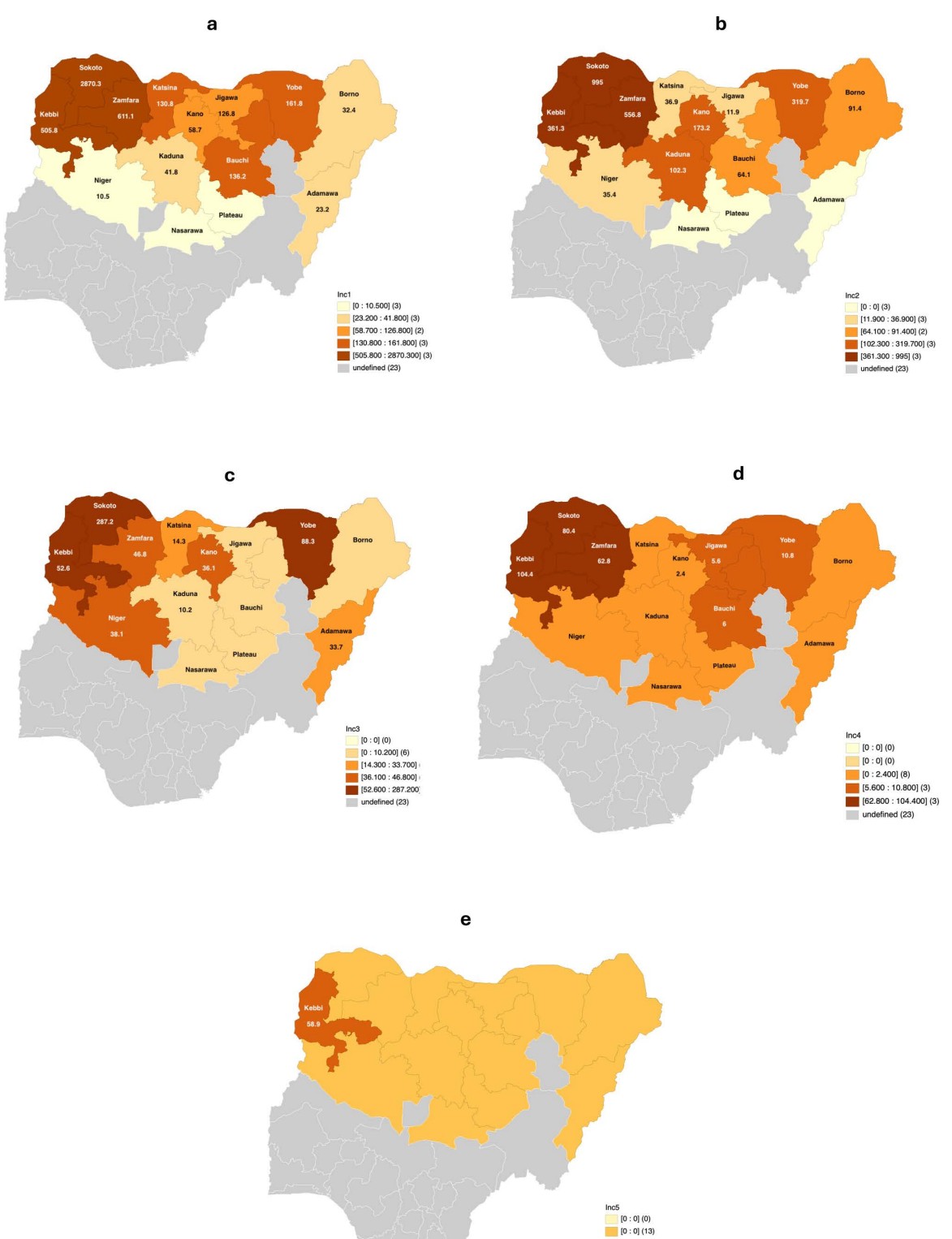

**Fig 6. Map showing the incidence of noma (per 100,000) population during the study period among different age groups (a) < 5 years (b) 5 – 9 years (c) 10 – 17 years (d) 18 – 39 years and (e) 40 – 64 years (Shapefile source:** https://data.humdata.org/dataset/cod-ab-nga**).**

## Conclusions

This study estimated the incidence of noma in northern Nigeria between 1999 and 2024 to be 87.8 cases per 100,000 population, with a slightly higher estimated incidence among males than females in the study period. The annual average and median estimated incidence of noma across all years was 3.4 and 1.6 cases per 100,000 (range: 0.2-16.6 cases per 100,000). Also, estimated incidence of noma was higher from 2020 to 2024 compared to all other years. Given that the recent increase in the estimated annual incidence may coincide with increased awareness in Northern Nigerian communities on noma and case presentation at NCH, additional studies are required to explain this finding.

Notably, we observed clustering of high incidence estimates in Sokoto, Kebbi, and Zamfara states, while Adamawa state had the lowest estimated incidence in the study period. However, this study found that the association between incidence and state of residence may be related to the individual's age, with Sokoto state having a high incidence of noma cases among those below 18 years, while Kebbi state had the highest estimated incidence of noma cases among those between young and middle-aged adults. These findings may guide stakeholders in the early identification of noma cases and provide knowledge of target populations for community sensitization among Northern Nigerian states that are represented in this incidence estimation study.

Our study also estimated that the incidence of gangrene in the study period, an irreversible stage of active noma disease, was significantly higher than cases with oedema or ANUG that are reversible. This finding showcases the need to continue intensified awareness efforts among rural communities in Northern Nigeria to improve case presentation of reversible noma disease stages. As poor access to treatment centers may also be implicated in the frequent irreversible disease presentation of noma cases in Northern Nigeria based on the NCH cohort, we suggest that relevant stakeholders set up a simple, fast, and possibly digital workflow connecting rural communities to the nearest treatment centers at the state, local government, or ward level to facilitate timely presentation and treatment of early noma disease.

## Acknowledgments

The authors wish to thank the entire staff of the Noma Children's Hospital, Sokoto for their efforts in providing comprehensive care for patients with noma in Nigeria. We also thank Peter Steinmann, Margaret Leila Srour, and Anaïs Galli for their insightful comments that helped improve the methods and reporting of this study.

## Author contributions

**Conceptualization:** Ramat Oyebunmi Braimah, John Adeoye.

**Data curation:** Ramat Oyebunmi Braimah, Abdurrazaq Olanrewaju Taiwo, Mujtaba Bala, Bruno Oludare Ile-Ogedengbe, Abubakar Abdullahi Bello.

**Formal analysis:** John Adeoye.

**Investigation:** Ramat Oyebunmi Braimah, John Adeoye, Mujtaba Bala.

**Methodology:** John Adeoye, Mujtaba Bala.

**Project administration:** Abubakar Abdullahi Bello.

**Resources:** Ramat Oyebunmi Braimah, Seidu Bello, Mujtaba Bala, Abubakar Abdullahi Bello.

**Validation:** John Adeoye, Abdurrazaq Olanrewaju Taiwo, Seidu Bello, Azeez Butali, Bruno Oludare Ile-Ogedengbe, Abubakar Abdullahi Bello.

**Visualization:** John Adeoye, Seidu Bello, Azeez Butali, Bruno Oludare Ile-Ogedengbe, Abubakar Abdullahi Bello.

**Writing – original draft:** Ramat Oyebunmi Braimah, John Adeoye.

**Writing – review & editing:** Ramat Oyebunmi Braimah, John Adeoye, Abdurrazaq Olanrewaju Taiwo, Seidu Bello, Mujtaba Bala, Azeez Butali, Bruno Oludare Ile-Ogedengbe, Abubakar Abdullahi Bello.

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
