## [Decision Letter · Decision Letter 0]

13 Mar 2025

PNTD-D-24-01910

Incidence and Clinical Presentation of Noma in Northern Nigeria (1999-2024)

Dear Dr. Adeoye,

Thank you for submitting your manuscript to PLOS Neglected Tropical Diseases. After careful consideration, we feel that it has merit but does not fully meet PLOS Neglected Tropical Diseases's publication criteria as it currently stands. Therefore, we invite you to submit a revised version of the manuscript that addresses the points raised during the review process.

Please submit your revised manuscript within 60 days May 12 2025 11:59PM. If you will need more time than this to complete your revisions, please reply to this message or contact the journal office at plosntds@plos.org. Please include the following items when submitting your revised manuscript:

We look forward to receiving your revised manuscript.

Kind regards,

Georgios Pappas

Section Editor

Georgios Pappas

Section Editor

Shaden Kamhawi

co-Editor-in-Chief

Paul Brindley

co-Editor-in-Chief

**Journal Requirements:**

1) <carina-action-element class="ng-star-inserted">Please ensure that the CRediT author contributions listed for every co-author are completed accurately and in full.</carina-action-element> 

<carina-action-element class="ng-star-inserted">At this stage, the following Authors/Authors require contributions: </carina-action-element><carina-action-element class="ng-star-inserted">Ramat Oyebunmi Braimah, John Adeoye, Abdurrazaq Olanrewaju Taiwo, Seidu Bello, Mujtaba Bala, Azeez Butali, Bruno Oludare Ile-Ogedengbe, and Abubakar Abdullahi Bello</carina-action-element><carina-action-element class="ng-star-inserted">. Please ensure that the full contributions of each author are acknowledged in the "Add/Edit/Remove Authors" section of our submission form.</carina-action-element> <carina-action-element class="ng-star-inserted">The list of CRediT author contributions may be found here: https://journals.plos.org/</carina-action-element><carina-action-element class="ng-star-inserted">plosntds</carina-action-element><carina-action-element class="ng-star-inserted">/s/authorship#loc-author-contributions</carina-action-element>  2) We ask that a manuscript source file is provided at Revision. Please upload your manuscript file as a .doc, .docx, .rtf or .tex. If you are providing a .tex file, please upload it under the item type u2018LaTeX Source Fileu2019 and leave your .pdf version as the item type u2018Manuscriptu2019.   3) <carina-action-element class="ng-star-inserted">Please upload all main figures as separate Figure files in .tif or .eps format. For more information about how to convert and format your figure files please see our guidelines: </carina-action-element> <carina-action-element class="ng-star-inserted">https://journals.plos.org/</carina-action-element><carina-action-element class="ng-star-inserted">plosntds</carina-action-element><carina-action-element class="ng-star-inserted">/s/figures</carina-action-element>  4) <carina-action-element class="ng-star-inserted">Some material included in your submission may be copyrighted. According to PLOSu2019s copyright policy, authors who use figures or other material (e.g., graphics, clipart, maps) from another author or copyright holder must demonstrate or obtain permission to publish this material under the Creative Commons Attribution 4.0 International (CC BY 4.0) License used by PLOS journals. Please closely review the details of PLOSu2019s copyright requirements here: PLOS Licenses and Copyright. If you need to request permissions from a copyright holder, you may use PLOS's Copyright Content Permission form.</carina-action-element> <carina-action-element class="ng-star-inserted">Please respond directly to this email and provide any known details concerning your material's license terms and permissions required for reuse, even if you have not yet obtained copyright permissions or are unsure of your material's copyright compatibility. Once you have responded and addressed all other outstanding technical requirements, you may resubmit your manuscript within Editorial Manager. </carina-action-element> <carina-action-element class="ng-star-inserted">Potential Copyright Issues:</carina-action-element> <carina-action-element class="ng-star-inserted">- Figures 3 and 6. Please provide a direct link to the base layer of the map (i.e., the country or region border shape) and ensure this is also included in the figure legend; and provide a link to the terms of use / license information for the base layer image or shapefile. We cannot publish proprietary or copyrighted maps (e.g. Google Maps, Mapquest) and the terms of use for your map base layer must be compatible with our CC BY 4.0 license. Note: if you created the map in a software program like R or ArcGIS, please locate and indicate the source of the basemap shapefile onto which data has been plotted. If your map was obtained from a copyrighted source please amend the figure so that the base map used is from an openly available source. Alternatively, please provide explicit written permission from the copyright holder granting you the right to publish the material under our CC BY 4.0 license. If you are unsure whether you can use a map or not, please do reach out and we will be able to help you. The following websites are good examples of where you can source open access or public domain maps: * U.S. Geological Survey (USGS) - All maps are in the public domain. (http://www.usgs.gov) * PlaniGlobe - All maps are published under a Creative Commons license so please cite u201cPlaniGlobe, http://www.planiglobe.com, CC BY 2.0u201d in the image credit after the caption. (http://www.planiglobe.com/?lang=enl) * Natural Earth - All maps are public domain. (http://www.naturalearthdata.com/about/terms-of-use/).</carina-action-element>

5) <carina-action-element class="ng-star-inserted">In the online submission form, you indicated that </carina-action-element><carina-action-element class="ng-star-inserted">"Data used for this study is not publicly available due to the need to maintain patient confidentiality. However, anonymized data may be obtained from the corresponding authors upon reasonable request"</carina-action-element><carina-action-element class="ng-star-inserted">. All PLOS journals now require all data underlying the findings described in their manuscript to be freely available to other researchers, either</carina-action-element> 

<carina-action-element class="ng-star-inserted">- In a public repository</carina-action-element> <carina-action-element class="ng-star-inserted">- Within the manuscript itself</carina-action-element> <carina-action-element class="ng-star-inserted">- Uploaded as supplementary information.</carina-action-element> <carina-action-element class="ng-star-inserted">This policy applies to all data except where public deposition would breach compliance with the protocol approved by your research ethics board. If your data cannot be made publicly available for ethical or legal reasons (e.g., public availability would compromise patient privacy), please explain your reasons by return email and your exemption request will be escalated to the editor for approval. Your exemption request will be handled independently and will not hold up the peer review process, but will need to be resolved should your manuscript be accepted for publication. One of the Editorial team will then be in touch if there are any issues.</carina-action-element> 

**Reviewers' Comments:**

Reviewer's Responses to Questions

**Key Review Criteria Required for Acceptance?**

**Methods:**

-Are the objectives of the study clearly articulated with a clear testable hypothesis stated?

-Is the study design appropriate to address the stated objectives?

-Is the population clearly described and appropriate for the hypothesis being tested?

-Is the sample size sufficient to ensure adequate power to address the hypothesis being tested?

-Were correct statistical analysis used to support conclusions?

-Are there concerns about ethical or regulatory requirements being met?

Reviewer #1: see comments

Reviewer #2: The methodology is well described and appropriate.

Reviewer #3: The objective of the study is mostly clear. There is no hypothesis, but as the objective is exploratory, I do not think it is necessary.

The study design has been adapted from previous studies (i.e. the WHO Delphi consultation). In my personal opinion, it would be interesting to try to calculate incidence by considering the population at risk, if the population data is available. However, if not the method can be used to a certain degree. One big limitation is that the formula is based on the proportion of surviving cases that have been referred. This number will differ greatly across states. For example, the proportion of surviving cases reaching the hospital in Sokoto who are originally from Sokoto will be much bigger compared to the proportion of cases reaching Sokoto from Adamawa state, which is on the other side of the country. Ideally, these different proportions should be estimated and adapted for the different states that are represented. If not, it should at least be critically reflected in the discussion as a major limitation. States cannot be compared as it is, otherwise we don’t know if the lower incidence the further a state is away from Sokoto is caused by the smaller population reaching Sokoto or a depiction of reality.

Another point where we need to be careful is that 2 states have been excluded because no noma cases have been reported. If we do this, we can not speak of an overall incidence across Northern Nigeria. By not including states without noma, we probably overestimate the incidence. If these states are not considered, the results need to be presented precisely and this issue needs to be discussed in the discussion.

**Results:**

-Does the analysis presented match the analysis plan?

-Are the results clearly and completely presented?

-Are the figures (Tables, Images) of sufficient quality for clarity?

Reviewer #1: see comments

Reviewer #2: The results are presented clearly. Figures and tables well presented

Reviewer #3: In general, the results are very comprehensive and interesting.

When reporting on age groups, I recommend to always differentiate between active and arrested noma (also in the text), otherwise we risk to confuse people about the age of noma onset. In general, in the demographic table, it would be nice to have a row of age of noma onset, if the data is available.

In the results tables, it is not always clear which comparisons the p-values concern. E.g. in table 1 we do not know if the p-value stems from a test comparing age groups or active and passive disease. If possible, it would be nice to see 95% Confidence intervals.

Language-wise, I recommend to be careful on how incidence and number of patients are presented. As it is written now, it might sometimes give the impression that the results represent the whole area or population groups, but the whole analysis is based on people who reached Sokoto noma hospital. For example, “the analysis showed that more patients had gangrene than oedema or ANUG” could be clearer like this “the analysis showed that more patients were admitted to the noma hospital with gangrene compared to oedema or ANUG”.

For incidence it is not always clear in the text if it concerns the study period or an average.

**Conclusions:**

-Are the conclusions supported by the data presented?

-Are the limitations of analysis clearly described?

-Do the authors discuss how these data can be helpful to advance our understanding of the topic under study?

-Is public health relevance addressed?

Reviewer #1: see comments

Reviewer #2: The conclusions are supported. I recommend discussion of limitations, which are not included. The public health relevance is addressed

Reviewer #3: The discussion could be less focused on the results but more critical about limitations. E.g. address difference in proportion of people reaching Sokoto depending on their geographic location, retrospective nature of study, is the study representative, do higher incidence rates in recent years really represent higher incidence or could they be caused by a bigger awareness of noma in Nigeria due to the engagement of the MoH and NGOs, consequently more people visit the Sokoto hospital, etc.

The conclusions mainly repeat the results but do not reflect on how this study advances our understanding of noma or its public health relevance. Even though it is a very relevant study with extremely valuable data.

**Editorial and Data Presentation Modifications?**

Reviewer #1: none

Reviewer #2: Please see in the general comments below.

Reviewer #3: 1. I highly discourage speaking of a "noma belt" or noma as an "African problem". We have most reports and noma data from the African region, but also most studies on noma were conducted there. Recent evidence suggests that noma is a global problem, wherever extreme poverty exists. By always focusing on the noma belt/the African continent, we discourage other regions to start looking for noma cases.

2. In figures 1&2 please clearly state that these are the noma cases registered in the noma hospital to avoid giving the impression that these are absolute and representative numbers.

3. Figures 5&6: you are speaking of noma incidence, hence referring to stages 1-3. In the results tables, the higher age groups did not present in stages 1-3, so I am not sure why they are presented in the incidence figures.

**Summary and General Comments:**

Reviewer #1: General:

- Braimah and co-authors reviewed a long case series of noma patients from the world’s premier noma hospital and tried to estimate incidence figures and morbidity variables stratified by state across northern Nigeria. This is an important study as few reliable estimates on noma exist, and even less depicting regional clustering and variations in morbidity indicators. The findings put forward by the team confirm some long-held assumptions, e.g. the link with poverty and age, but also add new elements such as the considerable number of older noma patients, some of them presumably presenting for treatment long after surviving noma.

A number of comments are offered for consideration in a further refined and revised version:

- general comment: the analysis is based on hospital data and as such, is inherently biased as only patients with access to the hospital could be included. The incidence estimates try to correct for under-reporting but this assumes homogenous access issues which clearly is not the case – distance certainly plays an important role. Also, no correction for morbidity indicators are used while presumably, the clinical picture of patients who died before reaching the hospital might have been different compared to the cohort that was included in the survey. The implications should be discussed in detail.

- abstract: at the beginning, noma is clearly described as a disease predominantly affecting young children while the results point to a considerable number of also older noma patients. This may be confusing and it should be made clear that acute noma primarily occurs in young children while arrested noma can affect individuals of any age.

- introduction: the geographic direction of east and west has been reversed in the description of noma distribution

- methods: should be written in past tense throughout the paragraph (e.g. involved instead of involves)

- methods: why was a clearly outdated census used as population reference?

- results: when mentioning incidence, it should always be made clear whether the incidence refers to a period of several years or it is per year.

- discussion: reasons for the sharp increase in noma cases over the past 5 years should be discussed as it follows a long period with relatively stable and much lower numbers

- no data on risk factors are presented beyond gender and age. Were they not available or analyzed? If available, at least selected indicators might be added as they may be important to understand temporal and regional dynamics.

Reviewer #2: Thank you to the authors for your research and manuscript on noma in northern Nigeria during the last two and a half decades. The title accurately describes the manuscript.

The authors performed a retrospective study of noma patients who presented to the Noma Children’s Hospital. Their study features the clinical presentation and estimates the incidence of noma in northern Nigeria. With the high and rising incidence of acute noma, the authors urge screening and treatment of acute noma throughout this region.

The abstract is well written. The first sentence could be revised. Noma is not a new tropical disease. As reported in the discussion section, noma is the most recent addition to the WHO list of Neglected Tropical Diseases (NTDs). Noma was included in a textbook of neglected diseases 400 years ago (Affectibus omissis’(Arnoldus Bootius, 1649, Neglected Diseases)

This research is limited to patients who presented to the Noma Children’s Hospital. Children with noma may not be recognized or able to travel to the hospital. Estimates are that only 10-30% of patients with acute noma receive treatment. The authors report that more patients presented in Stage 3 than in earlier stages, which may indicate late recognition or difficulties reaching the hospital. Due to the rapid progression of noma and high mortality, the true incidence may be much higher. Please discuss this limitation.

The last sentence in the introduction includes “genetic predisposition” as one of the noma risk factors. However, the references do not support this point, suggesting only that periodontal diseases may have genetic predispositions. Extreme poverty, chronic malnutrition, poor hygiene, underlying diseases or recent acute illness, and lack of dental care may compromise the patient’s immune system, resulting in microbial dysbiosis and susceptibility to noma.

Do the authors think the high incidence of noma cases in Sokoto is influenced by the presence of the Noma Children’s Hospital?

Regarding the last sentence of Data Collection, please correct the statement about acute and “arrested noma.” According to WHO stages, Stages 1 & 2 are reversible with treatment. Stages 3-5 are irreversible, and efforts are required to save the patient’s life. Even without treatment, some noma patients survive, but it is not known why their disease is arrested. Patients with “arrested noma” still progress through all the stages, including scarring and sequelae.

Thank you to the authors for your research on noma patients in Nigeria. This research should lead to a better understanding of this neglected disease and encourage effective interventions to eradicate this preventable childhood disease.

Reviewer #3: It is great to read about the incredible work that the noma hospital in Sokoto has been doing over the past years and very important to present data about the patient profiles. This information is invaluable and should definitely be published. An additional result that would be very interesting is the mortality rate of patients presenting with stages 1-3.

Also calculating the incidence of noma is very important. I am still hesitant about calculating a regional incidence based on hospital data from one state in the country. I doubt that with the current methodology it represents the incidence from states that are further away from Sokoto, because less patients will ever reach Sokoto. Therefore, please review my comments in the methods, results and conclusion boxes and consider:

1. Adapting the proportion of patients reaching the hospital based on geographic distance from Sokoto.

2. Re-integrating the states without any noma cases in your analysis or clearly discussing the implications of excluding them on your results.

3.Critically reflect your limitations in the discussion.

4. Rewrite the conclusion section to indicate how your study contributed to noma research and public health and what this means. E.g. do we need further studies, does this help us to better design interventions (e.g. most patients arrive at stage 3, how can we reach them in earlier stages), do you have recommendations.

PLOS authors have the option to publish the peer review history of their article (what does this mean? ). If published, this will include your full peer review and any attached files.

**Do you want your identity to be public for this peer review?** For information about this choice, including consent withdrawal, please see our Privacy Policy .

Reviewer #1: **Yes: ** Peter Steinmann

Reviewer #2: **Yes: ** Margaret Leila Srour

Reviewer #3: **Yes: ** Anaïs Galli

**Figure resubmission:**
---

## [Decision Letter · Decision Letter 1]

14 May 2025

Dear Dr. Adeoye,

We are pleased to inform you that your manuscript 'Estimated Incidence and Clinical Presentation of Noma in Northern Nigeria (1999-2024)' has been provisionally accepted for publication in PLOS Neglected Tropical Diseases.

Best regards,

Georgios Pappas

Section Editor

Georgios Pappas

Section Editor

Shaden Kamhawi

co-Editor-in-Chief

Paul Brindley

co-Editor-in-Chief

Reviewers are satsified with the current revision.

Reviewer's Responses to Questions

**Key Review Criteria Required for Acceptance?**

**Methods**

-Are the objectives of the study clearly articulated with a clear testable hypothesis stated?

-Is the study design appropriate to address the stated objectives?

-Is the population clearly described and appropriate for the hypothesis being tested?

-Is the sample size sufficient to ensure adequate power to address the hypothesis being tested?

-Were correct statistical analysis used to support conclusions?

-Are there concerns about ethical or regulatory requirements being met?

Reviewer #1: (No Response)

Reviewer #2: This revision is very clear about the objectives, study design, population description and statistical analysis.

**Results**

-Does the analysis presented match the analysis plan?

-Are the results clearly and completely presented?

-Are the figures (Tables, Images) of sufficient quality for clarity?

Reviewer #1: (No Response)

Reviewer #2: The results and figures are clear and well presented.

**Conclusions**

-Are the conclusions supported by the data presented?

-Are the limitations of analysis clearly described?

-Do the authors discuss how these data can be helpful to advance our understanding of the topic under study?

-Is public health relevance addressed?

Reviewer #1: (No Response)

Reviewer #2: The conclusions are well supported. The limitations are described adequately.

The authors discuss how the data is helpful and recommend additional studies.

The public health relevance is addressed.

**Editorial and Data Presentation Modifications?**

Reviewer #1: (No Response)

Reviewer #2: In the 8th line of the Results section, "most" is used to describe males which were only 54%. I suggest that more is appropriate for results less than 80-90%.

In the WHO Staging and Clinical presentation, the third line "most" presented with gangrene. I suggest "the majority" or "more" is appropriate.

**Summary and General Comments**

Reviewer #1: The authors have done a great job in considering all comments and revising the manuscript. No further comments from my side.

Reviewer #2: Thank you to the authors for responding to the reviewers. Your manuscript revision is well done and clearly represents the research you have accomplished. The incidence and clinical appearance of noma in northern Nigeria are well described. The relevance of your research and essential limitations can lead to further studies.

Thank you to the editors for your attention and support for this manuscript and for allowing my review of the original and revisions.

PLOS authors have the option to publish the peer review history of their article (what does this mean? ). If published, this will include your full peer review and any attached files.

**Do you want your identity to be public for this peer review?** For information about this choice, including consent withdrawal, please see our Privacy Policy .

Reviewer #1: **Yes: ** Peter Steinmann

Reviewer #2: **Yes: ** Margaret Leila Srour

---

## [Editor Report · Acceptance letter]

Dear Dr. Adeoye,

We are delighted to inform you that your manuscript, "Estimated Incidence and Clinical Presentation of Noma in Northern Nigeria (1999-2024)," has been formally accepted for publication in PLOS Neglected Tropical Diseases.

Best regards,

Shaden Kamhawi

co-Editor-in-Chief

Paul Brindley

co-Editor-in-Chief
